# Epitaxial growth of highly symmetrical branched noble metal-semiconductor heterostructures with efficient plasmon-induced hot-electron transfer

Li Zhai [1,2,3,14], Sara T. Gebre[4,14], Bo Chen [2,14], Dan Xu[1,14], Junze Chen[5], Zijian Li [2], Yawei Liu[4], Hua Yang[2], Chongyi Ling [2], Yiyao Ge[2], Wei Zhai[2], Changsheng Chen[6], Lu Ma[7], Qinghua Zhang[8], Xuefei Li[1], Yujie Yan[9], Xinyu Huang [10], Lujiang Li[2], Zhiqiang Guan[2], Chen-Lei Tao[1], Zhiqi Huang[2], Hongyi Wang[2], Jinze Liang[2], Ye Zhu[6], Chun-Sing Lee [2], Peng Wang[9,11], Chunfeng Zhang [10], Lin Gu[12], Yonghua Du [7], Tianquan Lian [4] ✉, Hua Zhang [2,3,13] ✉ & Xue-Jun Wu [1] ✉

Epitaxial growth is one of the most commonly used strategies to precisely tailor heterostructures with well-defined compositions, morphologies, crystal phases, and interfaces for various applications. However, as epitaxial growth requires a small interfacial lattice mismatch between the components, it remains a challenge for the epitaxial synthesis of heterostructures constructed by materials with large lattice mismatch and/or different chemical bonding, especially the noble metal-semiconductor heterostructures. Here, we develop a noble metal-seeded epitaxial growth strategy to prepare highly symmetrical noble metal-semiconductor branched heterostructures with desired spatial configurations, i.e., twenty CdS (or CdSe) nanorods epitaxially grown on twenty exposed (111) facets of Ag icosahedral nanocrystal, albeit a large lattice mismatch (more than 40%). Importantly, a high quantum yield (QY) of plasmon-induced hot-electron transferred from Ag to CdS was observed in epitaxial Ag-CdS icosapods (18.1%). This work demonstrates that epitaxial growth can be achieved in heterostructures composed of materials with large lattice mismatches. The constructed epitaxial noble metal-semiconductor interfaces could be an ideal platform for investigating the role of interfaces in various physicochemical processes.

Heterostructures have attracted tremendous attention due to their physicochemical properties arising from their morphologies[1-3], interfaces[4-6], and spatial arrangements of different components[7,8]. The precise control over their nanoscale structures is essential to understand the structure-property correlation and enhance their performance in various applications, such as electronics[9,10], catalysis[11,12], solar energy conversion[13,14], etc. To date, the seeded/templated epitaxial growth is the most commonly used strategy to precisely tailor the hierarchical heterostructures[15-19]. Epitaxial growth of a secondary material on a specific facet of the seed/template allows it to follow the crystallographic orientation of the seed/template[17,20]. Combined with delicate control over the exposed facet and crystal phase of seeds/

templates, this strategy has been successfully used to prepare various hierarchical heterostructures with well-defined spatial structures[21,22], interfaces[23,24], crystal phases[25–27], and programmable components[28,29].

However, the aforementioned seeded/templated epitaxial growth is mainly used to prepare heterostructures composed of components with similar lattice structures and/or chemical bonding, e.g., metal-metal[20,26,27], metal oxide-metal oxide[30,31], metal chalcogenide-metal chalcogenide[1,16,22,23,28], perovskite-perovskite[29], and metal-organic framework (MOF)-MOF[32,33] heterostructures due to the common requirement of similar structure (normally with lattice mismatch smaller than 5%) for the epitaxial growth[17,18,34]. Therefore, it remains a great challenge for the epitaxial growth of heterostructures made of distinctly different materials, especially for the noble metal-semiconductor heterostructures, in which two materials could have a large lattice mismatch (normally larger than 20%) and different chemical bonding[18,24,35–38]. Commonly, the preparation of noble metal-semiconductor heterostructures relies on conventional non-epitaxial strategies including chemical deposition[39,40], cation-exchange-facilitated non-epitaxial growth[35,41], phase transfer[42], photochemical deposition[43], chemical extraction[44,45], sol-gel method[46], etc. Although the obtained non-epitaxial noble metal-semiconductor heterostructures have shown great potential in diverse fields, such as catalysis[6,14,40,43,46], photovoltaic devices[4,14,47] and sensors[36,44,48], their wide application has been hindered due to difficulties in the precisely defined control over their structures and interfaces. Therefore, a reliable epitaxial growth method is urgently desired to rationally construct hierarchical noble metal-semiconductor heterostructures with precisely controlled architectures and well-defined interfaces.

In this work, we report a controlled synthesis of highly symmetrical branched noble metal-semiconductor heterostructures via a noble-metal-seeded epitaxial growth. The 1D II-VI semiconductor nanorods can be uniformly grown on the exposed (111) facets of zero-dimensional platonic Ag nanocrystals. The obtained heterostructure possesses the same crystallographic symmetry as the noble metal seed, i.e., twenty CdS (or CdSe) nanorods radially grown on twenty exposed (111) facets of the five-fold multi-twinned Ag icosahedron, respectively, denoted as Ag-CdS (or Ag-CdSe) icosapods. Importantly, the epitaxial growth of CdS (or CdSe) nanorods on Ag icosahedral nanocrystals has been achieved albeit a large lattice mismatch between Ag and CdS (or CdSe) (more than 40%). Impressively, the pump-probe transient absorption (TA) spectroscopy reveals the ultrafast plasmon hot-electron transfer process in the epitaxial heterostructures with high QY of the plasmon-induced hot-electron transferred from noble metal to semiconductor, i.e., 18.1% and 17.6% for the Ag-CdS and Ag-CdSe icosapods, respectively.

## Results

### Material synthesis and characterizations

Figure 1a schematically illustrates the synthesis of Ag-CdS icosapods. Briefly, five-fold multi-twinned Ag icosahedral nanocrystals with a size of 13.3 ± 0.6 nm were first synthesized (Supplementary Fig. 1), which were then used as seeds for the epitaxial growth of CdS nanorods via a hot injection method (see the Methods section). Scanning electron microscope (SEM) (Fig. 1b), transmission electron microscopy (TEM) (Fig. 1c) and high-angle annular dark-field scanning transmission electron microscopy (HAADF-STEM, Fig. 1d) images show the high purity and uniformity of the synthesized Ag-CdS icosapods with a size of 108.2 ± 5.5 nm, and the length and diameter of CdS nanorods are around 50 and 7 nm, respectively (Supplementary Fig. 2).

To reveal the detailed three-dimensional (3D) architecture of the obtained Ag-CdS icosapods, a series of HAADF-STEM images of a representative Ag-CdS icosapod tilted with different angles were collected. The 3D reconstruction shows that twenty CdS nanorods radially extend from twenty exposed facets of the Ag icosahedron seed (Supplementary Movie 1), confirming that the Ag-CdS icosapod has the

same rotational symmetry as the Ag icosahedral nanocrystal. As shown in Supplementary Fig. 3a–f, the Ag icosahedral nanocrystal exhibits twofold ($C_2$), threefold ($C_3$), and five-fold ($C_5$) symmetrical Moiré patterns, which are consistent with the previously reported five-fold multi-twinned icosahedral nanocrystals[49]. After the growth of CdS nanorods, the obtained Ag-CdS icosapods with $C_2$, $C_3$, $C_5$ symmetrical patterns (Supplementary Fig. 3g–i) have also been observed under HAADF-STEM (Fig. 1e–g), which are consistent with the HRTEM images (Supplementary Fig. 3j–l). In particular, when projected along its $C_3$ rotational axis (Fig. 1f), the Ag-CdS icosapod presents a sophisticated star-like pattern with the observed eighteen branches radially extended from the bright core, while the other two branches are unobserved since they are perpendicular to the projected plane. Moreover, a five-fold pattern of the Ag-CdS icosapod with the $C_5$ symmetry exhibits different contrast along the ten extending directions of branches, due to the overlapped images of two nanorods in the same direction when viewed along its $C_5$ rotational axis (Fig. 1g). The HAADF-STEM and the corresponding energy-dispersive X-ray spectroscopy (EDS) elemental mapping images (Fig. 1h) of a representative Ag-CdS icosapod reveal that the Ag element is located in the core, while the Cd and S elements are homogeneously distributed over the twenty nanorods. X-ray photoelectron spectroscopy (XPS) was used to determine the valence state of components in Ag-CdS icosapods (Supplementary Fig. 4). In particular, the peaks located at 373.7 and 367.7 eV can be ascribed to $Ag\ 3d_{3/2}$ and $Ag\ 3d_{5/2}$ of zerovalent Ag, respectively (Supplementary Fig. 4b). Combined XPS results with the Ag K-edge XANES spectra (Supplementary Fig. 5), it can be concluded that the Ag nanocrystals have not been sulfurized during the synthesis of Ag-CdS icosapods, which might arise from the trioctylphosphine (TOP) used in the synthesis[44,45,50]. Furthermore, the X-ray diffraction (XRD) pattern of the synthesized Ag-CdS icosapods can be well indexed to the wurtzite CdS and *fcc*-Ag (Supplementary Fig. 6). The aforementioned results clearly demonstrate the successful growth of CdS nanorods on the Ag icosahedral nanocrystals, and the obtained Ag-CdS icosapods show the uniform size as well as well-defined morphology with sophisticated symmetrical branched architecture. Ultraviolet photoelectron spectroscopy (UPS) was used to reveal the interfacial electronic structure of Ag-CdS icosapods. As shown in Supplementary Fig. 7a, the cut-off energies ($E_{cut-off}$) of Ag nanocrystals and Ag-CdS icosapods are determined to be 16.95 eV and 17.20 eV, respectively. The Fermi levels of Ag nanocrystals and Ag-CdS icosapods are estimated to be −4.27 eV and −4.02 eV relative to the vacuum level, respectively. Compared with the Ag nanocrystals, the shift (≈0.25 eV) of the Fermi level of Ag-CdS icosapods demonstrates a Schottky barrier was formed between Ag and n-type semiconductor CdS in the Ag-CdS icosapods[51]. Since the Fermi level, conduction band (CB), and valence band (VB) of CdS are measured to be −3.92 eV, −3.61 eV, and −6.22 eV, respectively (Supplementary Fig. 7b, c), the interfacial electronic structure of Ag-CdS icosapods can be schematically shown in Supplementary Fig. 7d.

Importantly, the synthesis of noble metal-semiconductor heterostructures with tunable components can be achieved. By exchanging the trioctylphosphine sulfide (TOPS) with trioctylphosphine selenium (TOPSe) as the chalcogen precursor in our experiment (see the "Methods" section), branched Ag-CdSe icosapods have been obtained. The TEM and HAADF-STEM images of Ag-CdSe icosapods (Supplementary Fig. 8) and the EDS elemental mapping result (Supplementary Fig. 9) of a representative Ag-CdSe icosapod demonstrate the successful growth of CdSe nanorods on the icosahedral Ag seeds. The XPS spectra of Ag-CdSe icosapods confirm the Ag core in Ag-CdSe icosapods maintained zerovalent (Supplementary Fig. 10), and the XRD spectrum matches well with the wurtzite CdSe and *fcc*-Ag (Supplementary Fig. 11).

### Epitaxial noble metal-semiconductor interfaces

The interfacial structures between the icosahedral Ag seed and the grown CdS and CdSe nanorods are very important for the formation of

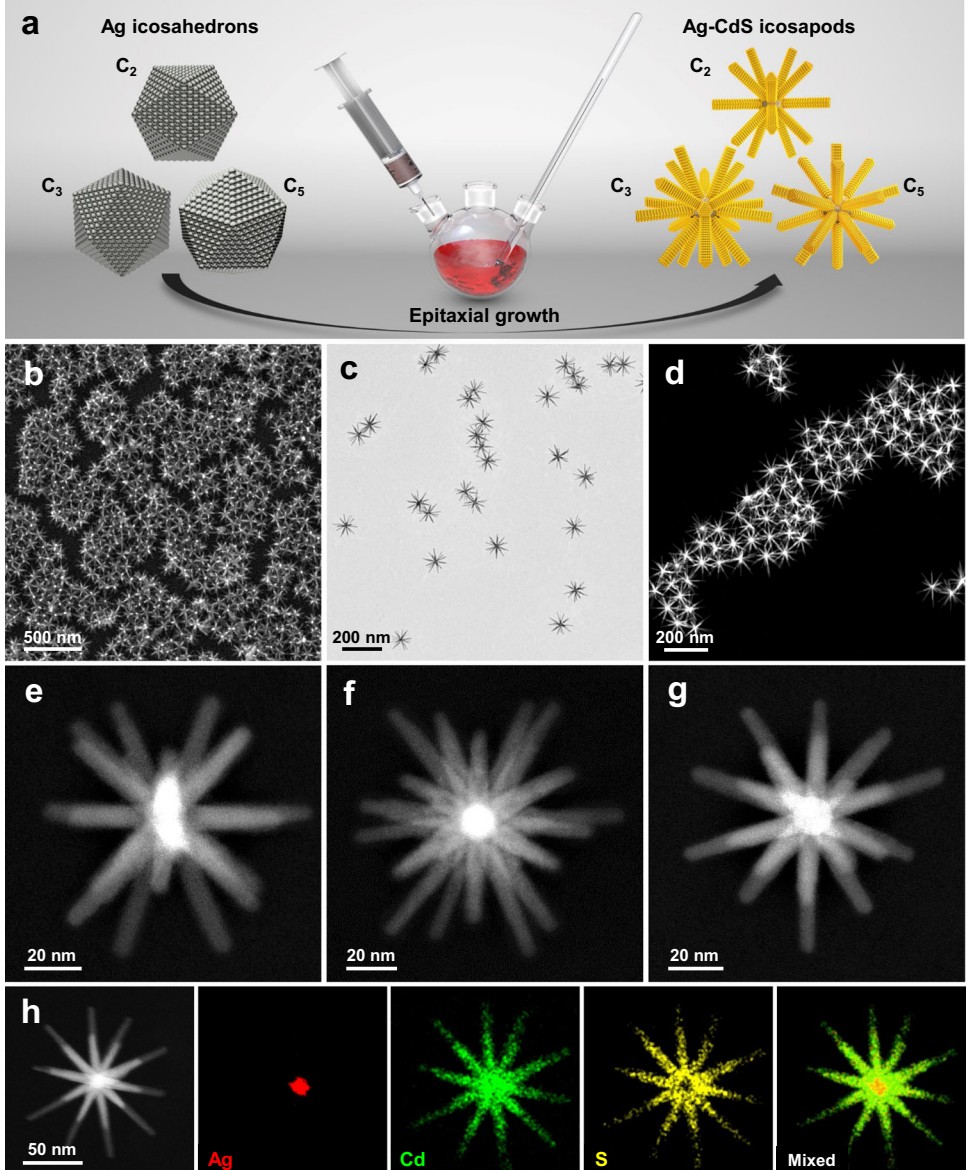

**Fig. 1 | Synthesis and characterization of Ag-CdS icosapods. a** Schematic illustration of the synthesis of Ag-CdS icosapods by using the pre-synthesized Ag nanocrystals, i.e., Ag icosahedral nanocrystals, as seeds for the epitaxial growth of CdS nanorods. $C_2$, $C_3$, and $C_5$ indicate the observed rotational axes of the schematically drawn Ag icosahedral nanocrystals (left) and Ag-CdS icosapods (right).

**b–d** SEM (**b**), TEM (**c**), and HAADF-STEM (**d**) images of Ag-CdS icosapods. **e–g** HAADF-STEM images of a representative Ag-CdS icosapod observed along its $C_2$ (**e**), $C_3$ (**f**), and $C_5$ (**g**) rotational axes, respectively. **h** HAADF-STEM and the corresponding elemental mapping images of a representative Ag-CdS icosapod.

such sophisticated noble metal-semiconductor heterostructures. As shown in Supplementary Fig. 12a, although it is very difficult to use spherical aberration-corrected high-angle annular dark-field STEM (Cs-HAADF-STEM) to clearly observe the detailed interfacial structure in the synthesized Ag-CdS icosapod due to its sophisticated architecture with large size of 108.2 ± 5.5 nm (Fig. 1b–d and Supplementary Fig. 2), the epitaxial growth of CdS rods on Ag with the epitaxial interface between the Ag (111) and CdS (002) planes, referred to as Ag (111)‖CdS (002), can be confirmed by the fast Fourier transformation (FFT) pattern (Supplementary Fig. 12b). To identify the Ag-CdS interface, Ag-CdS icosapods extracted from the reactor at ~60 s were characterized by the Cs-HAADF-STEM (Supplementary Fig. 13a). As shown in Fig. 2a, the measured lattice distances of 0.14, 0.24, 0.34 and 0.36 nm are well-matched with the interplanar distances of Ag (220), Ag (111), CdS (002), and CdS (100) planes, respectively. Besides the sharp Ag-CdS interface observed, it is also found that the Ag (111) planes are parallel to the CdS (002) planes, and Ag (220) planes are parallel to CdS (100) planes.

Together with the aligned spots in the FFT pattern (Fig. 2b), it can be concluded that the CdS are epitaxially grown on Ag icosahedral nanocrystals with an epitaxial relationship of Ag (111)‖CdS (002) and Ag (220)‖CdS (100). Moreover, as shown in the Cs-HAADF-STEM image (Fig. 2a) and inverse FFT image (Fig. 2c), the nearest neighbor atoms on the Ag surface are determined to be S of CdS (00$\bar{1}$) plane since the S atoms have lower contrast than the Cd atoms under the STEM mode, thus CdS is grown along the [001] direction[52] (Supplementary Fig. 14). Similarly, as shown in Fig. 2d–f, when CdSe was grown on the icosahedral Ag seeds at a short reaction time of 1 min (Supplementary Fig. 13b), instead of 3 min used for the preparation of aforementioned Ag-CdSe icosapods (Supplementary Figs. 8 and 9), an epitaxial relationship can also be found between Ag and CdSe. The epitaxial Ag-CdSe interface was determined to be Ag (111)‖CdSe (002) and Ag (220)‖CdSe (100), and CdSe grows along the [001] direction with the nearest neighbor Se atoms on the Ag surface (Fig. 2f). Based on the aforementioned results, the epitaxial interfaces of Ag-CdS and Ag-

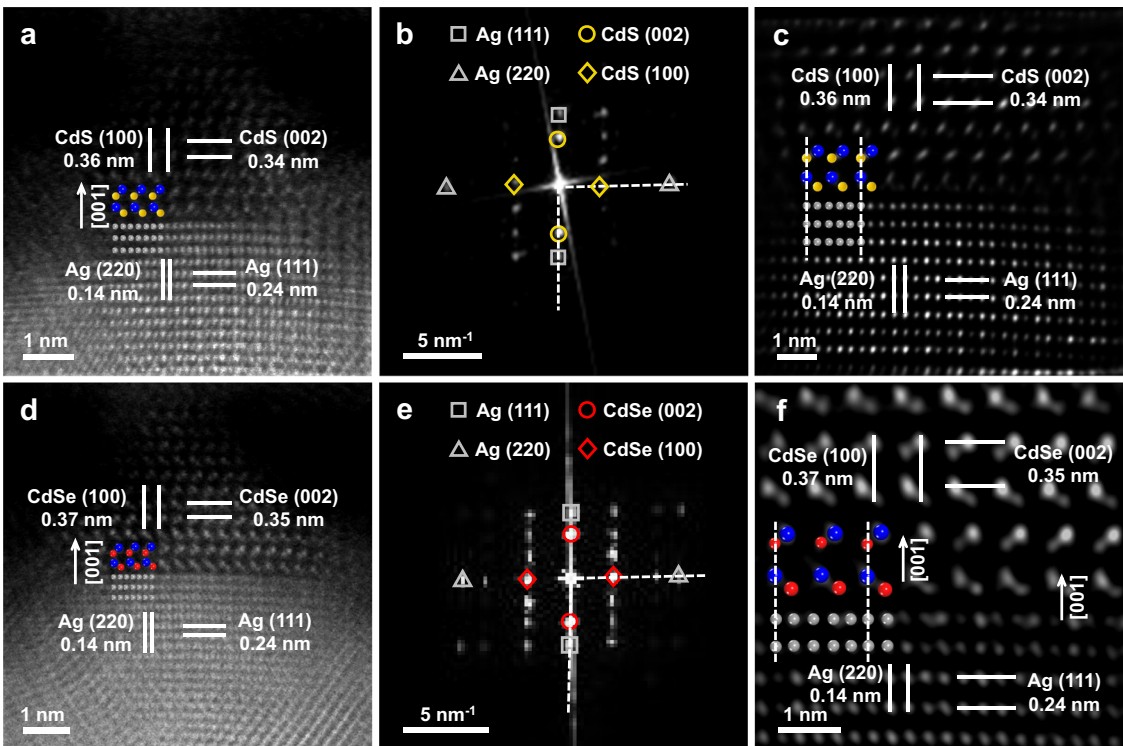

**Fig. 2 | Characterization of epitaxial interfaces in Ag-CdS and Ag-CdSe heterostructures obtained at a short reaction time of 1 min. a** Cs-HAADF-STEM image of the Ag-CdS interface. **b, c** The FFT pattern (**b**) and inverse FFT image (**c**) of the Ag-CdS interface. The silver square and silver triangle in (**b**) are applied to mark the FFT spots indexed to Ag (111) and Ag (220), respectively. The yellow circle and yellow rhombus are applied to mark the FFT spots indexed to CdS (002) and CdS (100), respectively. **d** Cs-HAADF-STEM image of the Ag-CdSe interface. **e, f** The FFT pattern (**e**) and inverse FFT image (**f**) of the Ag-CdSe interface. The silver square and silver triangle in (**e**) are applied to mark the FFT spots indexed to Ag (111) and Ag (220), respectively. The red circle and red rhombus are applied to mark the FFT spots indexed to CdSe (002) and CdSe (100), respectively. The silver, blue, yellow, red balls in (**a**), (**c**), (**d**), (**f**) represent the scheme of Ag, Cd, S, Se atoms, respectively.

CdSe can be schematically illustrated in Supplementary Fig. 15 and Supplementary Data 1. Importantly, despite the large lattice mismatch (more than 40%) between Ag and CdS (or CdSe) (Supplementary Table 1), which is much greater than the previously reported value for the epitaxial growth (normally <5%)[17,18,30,36], the epitaxial interfaces can still be achieved in our synthesized Ag-CdS and Ag-CdSe heterostructures.

## Ultrafast plasmon-induced charge transfer

The successful preparation of noble metal-semiconductor with well-defined morphologies and epitaxial interfaces provides a possibility to explore and understand their structure-property relationships. Here, we systematically studied the plasmon-induced charge transfer process in the obtained Ag-CdS and Ag-CdSe icosapods using pump-probe TA spectroscopy. As schematically shown in Fig. 3a, the pump laser-excited plasmon in Ag, which decays into electron-hole pairs, is followed by hot-electron transfer to semiconductor and/or direct interfacial charge transfer transition. The UV-visible absorption spectra of Ag icosahedral seeds, CdS nanorods (Supplementary Fig. 16), and the obtained Ag-CdS icosapods are presented in Fig. 3b. The strong peak for the blue curve located at 408 nm can be assigned to the localized surface plasmon resonance (LSPR) absorption peak of the Ag icosahedral nanocrystals. After the growth of CdS on Ag, two peaks located at 462 and 548 nm appear (red curve in Fig. 3b). The peak at 462 nm, which is similar to the absorption peak of CdS nanorods (465 nm, black curve in Fig. 3b), is assigned to the 1Σ exciton transition of CdS between the CB and valence band VB levels[53], and the peak at 548 nm should be attributed to the redshifted LSPR of Ag nanocrystals. We then performed pump-probe TA spectroscopy using the 550 nm laser to excite Ag-CdS icosapods. As the photon energy of 550 nm laser is below the

band gap of the CdS nanorods, the Ag LSPR bands in Ag-CdS icosapods are selectively excited (Supplementary Fig. 17). The TA spectrum of Ag-CdS icosapods, displayed as a two-dimensional pseudo-color plot in Fig. 3c, shows the distinct bleaching bands centered at ~480 and ~550 nm that can be attributed to the bleach of the CdS 1Σ band and Ag LSPR band, respectively. As the CdS was not excited, the bleach signal of the CdS 1Σ band can be attributed to the plasmon-induced hot-electron transfer from the Ag core to CdS nanorods.

Moreover, as shown in Fig. 3a, when the plasmon-induced hot-electron is transferred from the Ag to CdS or CdSe, it can generate a positive mid-IR TA signal due to the 1Σ-to-1Π intraband transition, which enables us to directly observe the plasmon-induced hot-electron transfer process and quantify its QY from Ag to CdS or CdSe[53–55]. As shown in the averaged mid-IR TA spectra of Ag-CdS icosapods (Fig. 3d), positive mid-IR TA signals were observed after the excitation of the Ag LSPR band, further confirming the presence of plasmon-induced hot-electron in the CB of CdS. The QY of plasmon-induced hot-electron transferred from Ag to CdS can be determined by the peak amplitude of the CdS intraband absorption signal according to Eq. 4, which is calculated to be 18.1% (see "Methods", Supplementary Figs. 18, 19 and Supplementary Table 2 for details). This QY value is much higher than that previously reported in nonepitaxial noble metal-semiconductor heterostructures[53,55–58], for instance, about 2.75% for Au-tipped CdS nanorods[53]. Fitting the time-resolved mid-IR absorption curve of Ag-CdS icosapods (Fig. 3e) based on Eq. 6 yielded a formation time of $18.0 \pm 0.3$ fs and a multi-exponential decay with a half-life of $1.9 \pm 0.16$ ps, which corresponds to plasmon-induced hot-electron transfer time and charge-recombination time, respectively (Supplementary Table 3). A similar result was obtained in the epitaxial Ag-CdSe icosapods. The measured QY and the time of plasmon-induced hot-

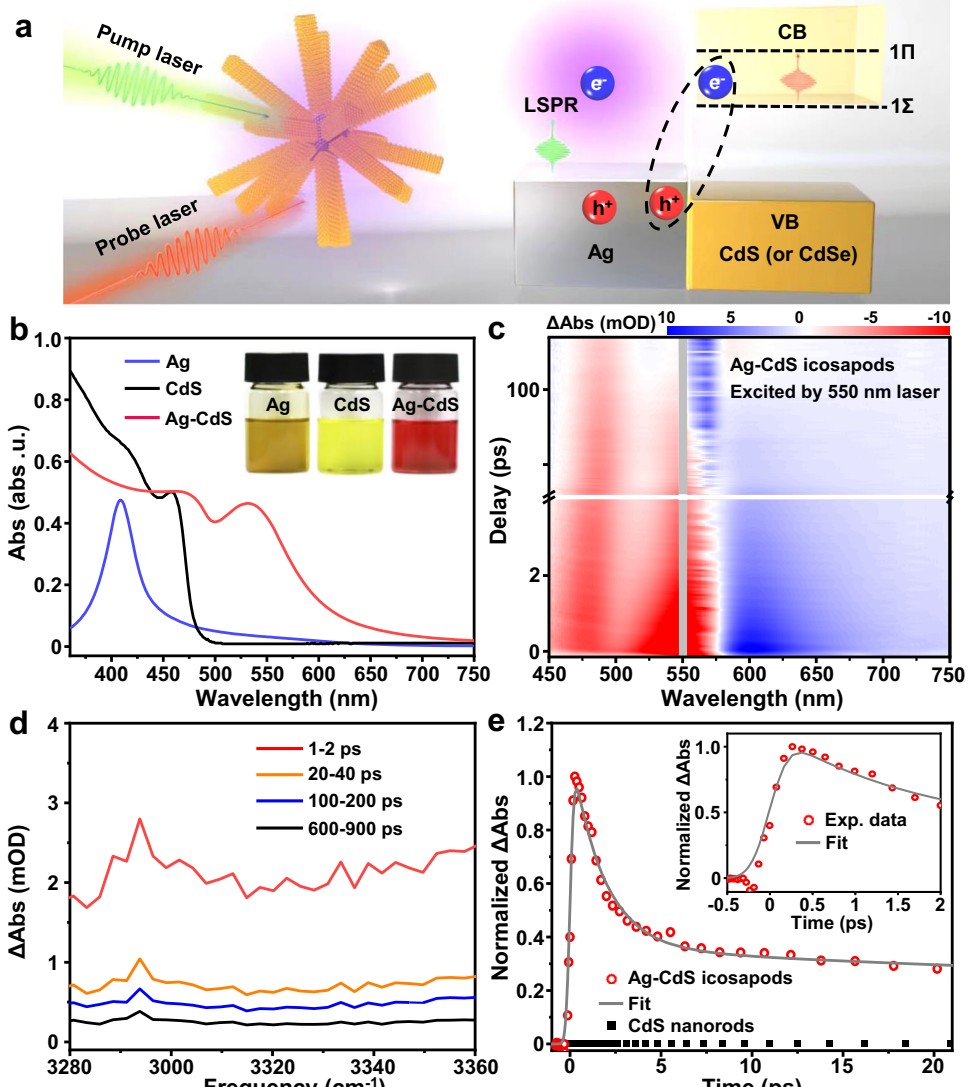

**Fig. 3 | Plasmon-induced charge separation in Ag-CdS icosapods. a** Schematic illustration of plasmon-induced hot-electron transfer in Ag-CdS icosapods after femtosecond laser pumping. The dashed ellipsis schematically illustrates the plasmon-induced interfacial charge-transfer transition in Ag-CdS icosapods. **b** UV-visible absorption spectra of the colloidal solutions of Ag icosahedral nanocrystals, CdS nanorods, and Ag-CdS icosapods in toluene. Insets: photographs of the corresponding colloid solutions. **c** Two-dimensional pseudo-color plots of pump-visible probe TA spectra of the colloid solutions of Ag-CdS in toluene. X-axis: probe wavelength; Y-axis: pump-probe delay; Color: change in absorbance (ΔAbs), shown

in milli-optical density unit (mOD). Gray shading is used to avoid the disturbance of the 550 nm pump laser to the probed signal. **d** Averaged mid-IR TA spectra of Ag-CdS icosapods from 1 to 2 ps (red curve), 20 to 40 ps (orange curve), 100 to 200 ps (blue curve), and 600 to 900 ps (black curve) after 550 nm laser excitation. **e** Time-resolved mid-IR absorption spectrum of Ag-CdS icosapods. Gray curves represent the multiexponential fits as described in Eq. 6. A negligible intraband absorption signal (black squares) is observed in a control experiment conducted on CdS nanorods. Inset in **e**: zoom-in kinetics spectrum for the Ag-CdS icosapods. Source data are provided as a Source data file.

electron transfer were determined to be 17.6% and 7.5 ± 7.2 fs when epitaxial Ag-CdSe icosapods were pumped by a 700 nm laser, respectively (Supplementary Figs. 19–22; Supplementary Tables 3 and 4). The plasmon-induced hot-electron transfer processes in the epitaxial Ag-CdS and Ag-CdSe icosapods are much faster than the conventional indirect plasmon-induced hot-electron transfer process (commonly more than 100 fs[53,55]). On such a short timescale (<20 fs), the plasmon directly decays into the electrons in the CB of semiconductors via the interfacial charge-transfer transition before rapid cooling of hot electrons induced by electron-electron scattering[54,55,58–60]. Unfortunately, because the transient IR selectively probes the electrons injected into the CB of CdS, similar to the previous reports[54], our study was not able to provide direct information about the effect of the metal-semiconductor interfacial states on the plasmon hot electron transfer and recombination processes.

In summary, we have demonstrated a noble-metal-seeded epitaxial growth method to achieve the controlled synthesis of highly symmetrical noble metal-semiconductor heterostructures, i.e., Ag-CdS and Ag-CdSe icosapods. Abrupt epitaxial Ag-CdS/CdSe interfaces are observed in the Ag-CdS and Ag-CdSe icosapods, which could be concluded as Ag (111)‖CdS/CdSe (002) and Ag (220)‖CdS/CdSe (100), albeit a large lattice mismatch (more than 40%). Detailed pump-probe TA-IR spectroscopy studies reveal high QYs of plasmon-induced hot-electron transferred from noble metal to semiconductor in the Ag-CdS (18.1%) and Ag-CdSe (17.6%) icosapods. This work demonstrates a strategy to rationally design and controllably synthesize sophisticated hierarchical noble metal-semiconductor heterostructures. The constructed epitaxial interface provides an ideal platform for investigating the role of the metal-semiconductor interface in various physico-chemical processes.

## Methods

### Materials

Silver nitrate (AgNO$_3$, 99.9999% trace metals basis), cadmium oxide (CdO, ≥99.99% trace metals basis), cadmium chloride (CdCl$_2$, 99.99% trace metals basis), sulfur powder (S, ≥99.98% trace metals basis), selenium powder (Se, 99.99%), oleylamine (technical grade, 70%), trioctylphosphine (97%), trioctylphosphine oxide (technical grade, 90%), oleic acid (technical grade, 90%) were purchased from Sigma-Aldrich. N-octadecylphosphonic acid (95%), n-hexylphosphonic acid (95%) were bought from Alfa Aesar. Hexane (≥99.0%), toluene (≥99.0%), and methanol (≥99.9%) were purchased from Aladdin and used as received without any further purification.

### Synthesis of five-fold multi-twinned Ag icosahedral nanocrystals

Typically, after 85 mg of AgNO$_3$ were mixed with 10 mL of oleylamine in a 50 mL three-neck flask, the mixture was degassed at 60 °C for 30 min under vigorous magnetic stirring at 750 r.p.m. It was then quickly heated up (≥15 °C min$^{-1}$) to 220 °C under N$_2$ atmosphere. After the temperature was maintained at 220 °C for 1 h, the solution was cooled down to 160 °C, and a mixture of 60 μL of acetate and 0.5 mL of oleylamine was injected into the flask. After the temperature was kept at 160 °C for 10 min, the reaction was stopped by removing the heating mantle. The flask was then cooled down to room temperature naturally. The obtained brownish product was precipitated by centrifuge at 5800 × g. for 2 min. The precipitate was then washed 3 times using the mixture of hexane and acetone (V$_{hexane}$/V$_{acetone}$ = 5/1). Finally, the obtained Ag nanocrystals were re-dispersed in 2.5 mL of TOP as seeds for the synthesis of Ag-CdS icosapods and Ag-CdSe icosapods.

### Synthesis of Ag-CdS and Ag-CdSe icosapods

TOPS and TOPSe stock solutions were first prepared by dissolving 18 mg of sulfur powder or 10 mg of selenium powder in 0.5 mL of TOP in a glove box filled with argon, respectively. For the synthesis of Ag-CdS icosapods, 115 mg of CdO, 18 mg of CdCl$_2$, 580 mg of n-octadecylphosphonic acid, 160 mg of n-hexylphosphonic acid, and 6 g of trioctylphosphine oxide were added into a 50 mL three-neck flask and degassed at 120 °C for 30 min under vigorous magnetic stirring at 750 r.p.m. After it was heated to 360 °C under N$_2$ atmosphere, the solution becomes transparent. Then, 1 mL of TOP was injected into the flask, and the temperature was kept at 360 °C for 15 min. A mixture of 0.5 mL of the above-mentioned Ag seeds in TOP and 0.5 mL of TOPS stock was injected into the flask at 360 °C swiftly. After the injection of Ag seeds and TOPS, the temperature was kept at 350 °C for 6 min. Then the reaction was stopped by removing the heating mantle. After the solution was cooled down to 100 °C, 10 mL of toluene was injected into the reaction flask. Then, 5 mL of methanol was added to the solution, and the product was collected by centrifuge at 5800 × g for 3 min. The obtained precipitate was washed with the mixture of toluene and methanol (V$_{toluene}$/V$_{methanol}$ = 5/1) and then dispersed into 50 mL of toluene. For the synthesis of Ag-CdSe icosapods, all the conditions are the same with the synthesis of Ag-CdS icosapods except the mixture of 0.5 mL of the above-mentioned Ag seeds in TOP and 0.5 mL of TOPSe stock was injected into the flask at 310 °C and the temperature was kept at 300 °C for 3 min. Note that the narrow size distribution of Ag seeds (Supplementary Fig. 1) is critical for synthesizing such delicate Ag-CdS and Ag-CdSe icosapods.

### Synthesis of CdS and CdSe nanorods

TOPS and TOPSe stock solutions were first prepared by dissolving 32 mg of sulfur powder or 40 mg of selenium powder in 0.5 mL of TOP in a glove box filled with argon, respectively. For the synthesis of CdS nanorods, 120 mg of CdO, 580 mg of n-octadecylphosphonic acid, 160 mg of n-hexylphosphonic acid, and 6 g of trioctylphosphine oxide were added into a 50 mL three-neck flask and degassed at 120 °C for 30 min under vigorous magnetic stirring at 750 r.p.m. After it was

heated to 370 °C under N$_2$ atmosphere, the solution becomes transparent. Then, 1 mL of TOP was injected into the flask, and the temperature was kept at 360 °C for 15 min. 0.5 mL of TOPS stock was injected into the flask at 370 °C swiftly, and the temperature was kept at 360 °C for 3 min. Then, the reaction was stopped by removing the heating mantle. After the solution was cooled down to 100 °C, 10 mL of toluene was injected into the reaction flask. Then, 5 mL of methanol was added to the solution, and the product was collected by centrifuge at 5800 × g for 3 min. The obtained precipitate was washed with the mixture of toluene and methanol (V$_{toluene}$/V$_{methanol}$ = 5/1) and then dispersed into 50 mL of toluene. For the synthesis of CdSe nanorods, all the conditions are the same as the synthesis of CdS nanorods except 0.5 mL of TOPSe stock was injected into the flask at 380 °C and the temperature was kept at 360 °C for 1 min.

### Characterization

Powder XRD patterns of the dried samples were recorded on Bruker D8 diffractometer at a scanning rate of 2° min$^{-1}$ using Cu Kα radiation (λ = 1.5406 Å). XPS and UPS spectra were recorded on the ESCALAB 250Xi (Thermo Fisher Scientific) system. The XPS results were calibrated with the reference C 1s peak located at 284.6 eV. The X-ray absorption fine structure (XAFS) experiments under transmission mode were performed at the 7-BM of the National Synchrotron Light Source II (NSLS-II) which is using a channel-cut monochromator. The NSLS-II is rung at 400 mA electron beam current under top-off mode. Samples for SEM and TEM characterizations were prepared by dropping nanomaterial dispersions in toluene on a silicon substrate and amorphous carbon-coated copper grid, respectively. SEM images were obtained using FESEM, JEOL JSM-7800F. TEM characterization was performed with a JEOL JEM-2100 and FEI TF20 operated at 200 kV. Dark-field STEM, Cs-HAADF STEM, and elemental mapping images were obtained using FEI Titan G2 60-300 operated at 300 kV as well as a JEM-ARM200F operated at 200 kV. The 3D reconstruction of Ag-CdS icosapods was carried out using FEI Titan G2 60-300 operated at 60 kV.

### Visible femtosecond transient absorption

The visible femtosecond transient absorption measurements were conducted in a Helios spectrometer (Ultrafast Systems LLC) with pump and probe beams derived from a regenerative amplified Ti: Sapphire laser system (Coherent Astrella, 35 fs, 4 mJ per pulse, and 1 kHz repetition rate). The 800 nm output pulse was split into two beams with a beam splitter. One beam passed through a tunable optical parametric amplifier (OperA solo, Coherent) to generate a tunable visible pump. During the measurement, the pump beam was chopped by a synchronized chopper to 500 Hz. The other beam was attenuated and focused on a CaF$_2$ window to generate the white light continuum with a wavelength range from 350 nm to 800 nm, referred to as the probe beam. The probe beam was focused into a 1-mm pathlength quartz cuvette (Starna) containing the sample in toluene. The transmission of the probe was collected by a fiber optics-coupled multichannel spectrometer with complementary metal-oxide-semiconductor (CMOS) sensors and detected at a frequency of 1 kHz (Ultrafast systems, Helios). The delay between the pump and probe pulses was controlled by a motorized delay stage. Samples in 1-mm cuvettes were used for all spectroscopy measurements and stirred vigorously during the measurements. The data were analyzed with surface Xplorer.

### Mid-IR femtosecond transient absorption

Mid-IR TA experiments were conducted using a commercial Ti:Sapphire regenerative amplifier (Astrella, Coherent) at 800 nm with a repetition rate of 1 kHz and pulse duration of 35 fs. An optical amplifier (OperA solo, Coherent) pumped by the regenerative amplifier was used to provide a pump beam with tunable wavelengths. Another

optical amplifier (OperA solo, Coherent) pumped by the regenerative amplifier was used to generate a tunable IR probe via difference frequency generation (DFG) by signal and idler beams. The pump beam was chopped by a synchronized chopper to 500 Hz and focused on a 1.5-mm path-length Harrick cell containing the sample in toluene. The IR probe, which was also focused on a 1.5-mm path length Harrick cell containing the sample in toluene, overlapped with the temporally delayed pump beam controlled by a movable delay stage. The probe intensity with and without the pump was analyzed by a Teledyne/ phasetech nitrogen-cooled $128 \times 128$ MCT detector. All measurements were carried out under ambient conditions.

### QY measurement of the plasmon-induced hot-electron transfer in Ag-CdS heterostructures

The QY of the plasmon-induced hot-electrons transferred from the Ag core to the CdS under 550 nm excitation can be quantified by the mid-IR intraband absorption at around 3340 $cm^{-1}$ of CB electrons in CdS. The CB electron in CdS produces a broad IR absorption which can be attributed to the transition from $1\Sigma$ to a higher level (such as $1\Pi$), as shown in Fig. 3a. To determine the plasmon-induced hot-electron transfer QY, the amplitude of IR signal per CdS CB electron was quantified, denoted as $S_0$. Hence, $S_0$ can be calculated by Eq. 1:

$$S_0 = \frac{\Delta A(\text{CdS}, 400 \text{ nm})}{N(\text{CdSCBelectrons})} \quad (1)$$

where $\Delta A$ is the maximum signal amplitude obtained from the kinetics; N(CdS CB electrons) is the number of electrons in CdS CB band, which can be determined by the number of absorbed photons by CdS (every photon creates one electron in CdS CB band under 400 nm excitation). Therefore, N(CdS CB electrons) equals the pump photon flux in the pump/probe overlap region times the sample absorption $\left( \frac{\text{power}/h\nu}{\text{beam size}} \times (1 - 10^{-\text{OD}}) \right)$, where $h\nu$ is photon energy.

The QY is calculated by the number of hot electrons generated in Ag-CdS per absorbed photons under 550 nm excitation, as shown in Eq. 2.

$$
\begin{aligned}
QY &= \frac{N(\text{hot electrons})}{N(\text{absorbed photons})} \\
&= \frac{\frac{\Delta A(\text{Ag}-\text{CdS},550\text{ nm})}{S_0}}{\frac{\text{power}(550\text{ nm})/h\nu(550\text{ nm})}{\text{beam size}(550\text{ nm})} \times \left(1 - 10^{-\text{OD}(\text{Ag}-\text{CdS},550\text{ nm})}\right)} \\
&= \frac{\text{beam size}(550\text{ nm})}{\text{beam size}(400\text{ nm})} \times \left[ \frac{\frac{\Delta A(\text{Ag}-\text{CdS},550\text{ nm})}{\text{power}(550\text{ nm})/h\nu(550\text{ nm})}}{\left(1 - 10^{-\text{OD}(\text{Ag}-\text{CdS},550\text{ nm})}\right)} \Big/ \frac{\frac{\Delta A(\text{CdS},400\text{ nm})}{\text{power}(400\text{ nm})/h\nu(400\text{ nm})}}{\left(1 - 10^{-\text{OD}(\text{CdS},400\text{ nm})}\right)} \right]
\end{aligned}
$$

$$(2)$$

In Eq. 2, beam size (550 nm) and beam size (400 nm) represent the overlap areas of pump and probe beams under 550 nm, and 400 nm pump laser excitation, respectively, whose ratio can be determined in a calibration sample ($Cd_3P_2$ quantum dots in this work) under the same conditions. The calibration sample is excited at both wavelengths and is expected to have the same QY for generating CB electrons. Then $\frac{\text{beam size}(550\text{ nm})}{\text{beam size}(400\text{ nm})}$ can be described as follows:

$$\frac{\text{beam size}(550\text{ nm})}{\text{beam size}(400\text{ nm})} = \frac{\frac{\Delta A(\text{Cd3P2},400\text{ nm})}{\text{power}(400\text{ nm})/h\nu(400\text{ nm})}}{\left(1 - 10^{-\text{OD}(\text{Cd3P2},400\text{ nm})}\right)} \Big/ \frac{\frac{\Delta A(\text{Cd3P2},550\text{ nm})}{\text{power}(550\text{ nm})/h\nu(550\text{ nm})}}{\left(1 - 10^{-\text{OD}(\text{Cd3P2},550\text{ nm})}\right)} \quad (3)$$

Thus, the QY of plasmon-induced hot-electron in Ag-CdS icosapods is given by Eq. 4.

$$QY = \frac{\frac{\Delta S(\text{Ag}-\text{CdS},550\text{ nm})}{1-10^{-\text{OD}(\text{Ag}-\text{CdS},550\text{ nm})}}}{\frac{\Delta S(\text{CdS},400\text{ nm})}{1-10^{-\text{OD}(\text{CdS},400\text{ nm})}}} \times \frac{\frac{\Delta S(\text{Cd3P2},400\text{ nm})}{1-10^{-\text{OD}(\text{Cd3P2},400\text{ nm})}}}{\frac{\Delta S(\text{Cd3P2},550\text{ nm})}{1-10^{-\text{OD}(\text{Cd3P2},550\text{ nm})}}} \quad (4)$$

Here, we define $\Delta S = \frac{\Delta A}{\text{power}}$, a quantity to be obtained from the transient IR measurement. To improve the measurement accuracy and to check the linearity of the signal, $\Delta A$ as a function of excitation power was measured for all samples. As shown in Supplementary Fig. 18, the TA signal amplitude increases linearly with the excitation power. $\Delta S$ can be obtained from the slope of the fitted linear line. For all samples, the OD is controlled at around 0.2–0.8. The values of epitaxial Ag-CdS icosapods are listed in Supplementary Table 2. It should be noted that because the nanorods are directly grown on the Ag nanoparticles, there may be slight differences in the diameter of the CdS nanorods control sample compared to the diameter of the nanorods in the heterostructure. This could affect the QY as the optical properties of the nanorods are dependent upon their size (absorption cross section), therefore, the QYs for each sample were calculated to the best of our ability using the independently synthesized CdS nanorods controls.

### QY measurement of the plasmon-induced hot-electron transfer in Ag-CdSe icosapods

The QY of the electrons transferred from the Ag core to the CdSe under 700 nm excitation can be quantified by the mid-IR intraband absorption at around 3340 $cm^{-1}$ of CB electrons in CdSe. Similar to Ag-CdS icosapods, the QY is calculated by the number of hot electrons generated in Ag-CdSe icosapods per absorbed photons under 700 nm excitation. CdSe nanorods are excited by a 400 nm laser. $Cd_3P_2$ quantum dots are excited by 400 nm laser and 700 nm laser for calibration. The QY of plasmon-induced hot-electron transfer in Ag-CdSe icosapods was shown in Eq. 5.

$$QY = \frac{\frac{\Delta S(\text{Ag}-\text{CdSe},700\text{ nm})}{1-10^{-\text{OD}(\text{Ag}-\text{CdSe},700\text{ nm})}}}{\frac{\Delta S(\text{CdSe},400\text{ nm})}{1-10^{-\text{OD}(\text{CdSe},400\text{ nm})}}} \times \frac{\frac{\Delta S(\text{Cd3P2},400\text{ nm})}{1-10^{-\text{OD}(\text{Cd3P2},400\text{ nm})}}}{\frac{\Delta S(\text{Cd3P2},700\text{ nm})}{1-10^{-\text{OD}(\text{Cd3P2},700\text{ nm})}}} \quad (5)$$

For all samples, the OD is controlled at around 0.4–0.7. The values used in Eq. 4 is listed in Supplementary Table 4.

### The multiexponential fit of the decay kinetics of Ag-CdS and Ag-CdSe icospads

The normalized TA IR kinetics shown in Fig. 3e and Supplementary Fig. 23b are fitted by a single exponential rise and multi-exponential decay function according to Eq. 6:

$$S(t) = \text{IRF} \otimes \left[ -\exp\left(-\frac{t}{T_f}\right) + \sum_{1}^{3} A_i \exp\left(-\frac{t}{T_i}\right) \right] \quad (6)$$

The instrument response function (IRF) of the measurement was determined in a 1 mm silicon wafer and can be well modeled by a Gaussian function with an FWHM of -177 ± 25 fs. $T_f$ is the time constant of the signal rise. $A_i$ and $T_i$ ($i = 1,2,3$) are amplitude and time constants of the multiexponential function decay function, respectively, $\sum_{1}^{3} A_i = 1$. The fitting parameters are listed in Supplementary Table 3.

### Reporting summary

Further information on research design is available in the Nature Portfolio Reporting Summary linked to this article.

## Data availability

The raw data that support the results of this study are available from the corresponding authors on request. Source data are provided with this paper.

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

## Acknowledgements

This work was supported by the National Natural Science Foundation of China (21871129 and 22271142), the Fundamental Research Funds for the Central Universities (020514380204), "Innovation & Entrepreneurship Talents Plan" of Jiangsu Province, and start-up funds from Nanjing University. H.Z. thanks the support from ITC via Hong Kong Branch of National Precious Metals Material Engineering Research Center (NPMM), and the Start-Up Grant (Project No. 9380100), and the grants (Project Nos. 9678272 and 1886921) from the City University of Hong Kong. T.L. acknowledges the support from the U.S. Department of Energy, Office of Science, Office of Basic Energy Sciences, Solar Photochemistry Program under Award Number (DE-SC0008798). S.T.G. acknowledges support from an AGEP supplement to NSF award number CHE-2004080. L.G. thanks the Beijing Natural Science Foundation (Z190010), National Natural Science Foundation of China (51991344, 52072400, 52025025). This research used 7-BM and 8-BM of the National Synchrotron Light Source II, a U.S. Department of Energy (DOE) Office of Science User Facility operated for the DOE Office of Science by Brookhaven National Laboratory under Contract No. DE-SC0012704.

## Author contributions

H.Z. and X.-J.W. proposed the research direction. H.Z., X.-J.W., and T.L. supervised the project. L.Z., D.X., and J.C. carried out the synthesis of the materials. L.Z., Z.L., and D.X. performed HRTEM measurements. B.C., L.Z., Y.Y., C.C., Q.Z., Y.Z., P.W., and L.G. collected dark-field STEM and EDS elemental mapping data. L.Z., S.T.G., Y.L., X.H., and C.Z. performed steady-state and TA measurements. C.L. built the 3D model of the interface. H.Y., W.Z., X.L., C.-L.T., Z.G., and C.-S.L. contributed to XPS, SEM, and XRD characterization. L.M. and Y.D. analyzed synchrotron data. Y.G., L.L., Z.H., H.W., Y.D., and J.L. discussed the experimental results. L.Z., S.T.G., Y.L., T.L., X.-J.W., and H.Z. wrote the manuscript. All authors checked the manuscript and agreed with its content.

## Competing interests

## Additional information

[1]State Key Laboratory of Coordination Chemistry, School of Chemistry and Chemical Engineering, Nanjing University, Nanjing 210023, China. [2]Department of Chemistry, City University of Hong Kong, Hong Kong, China. [3]Hong Kong Branch of National Precious Metals Material Engineering Research Center (NPMM), City University of Hong Kong, Hong Kong, China. [4]Department of Chemistry, Emory University, Atlanta, GA 30322, USA. [5]College of Materials Science and Engineering, Sichuan University, Chengdu, Sichuan 610065, China. [6]Department of Applied Physics, The Hong Kong Polytechnic University, Hung Hom, Hong Kong. [7]National Synchrotron Light Source II, Brookhaven National Laboratory, Upton, NY 11973, USA. [8]Beijing National Laboratory for Condensed Matter Physics, Institute of Physics, Chinese Academy of Sciences, Beijing 100190, China. [9]National Laboratory of Solid State Microstructures, Jiangsu Key Laboratory of Artificial Functional Materials, College of Engineering and Applied Sciences and Collaborative Innovation Center of Advanced Microstructures,

Nanjing University, Nanjing 210093, China. [10]National Laboratory of Solid State Microstructures, School of Physics, and Collaborative Innovation Center of Advanced Microstructures, Nanjing University, Nanjing 210093, China. [11]Department of Physics, University of Warwick, Coventry CV4 7AL, UK. [12]Beijing National Center for Electron Microscopy and Laboratory of Advanced Materials, Department of Materials Science and Engineering, Tsinghua University, Beijing 100084, China. [13]Shenzhen Research Institute, City University of Hong Kong, Shenzhen 518057, China. [14]These authors contributed equally: Li Zhai, Sara T. Gebre, Bo Chen, Dan Xu. ✉e-mail: tlian@emory.edu; hua.zhang@cityu.edu.hk; xjwu@nju.edu.cn

