## [Peer Review File · Nature Communications]

Epitaxial Growth of Highly Symmetrical Branched Noble Metal-Semiconductor Heterostructures with Efficient Plasmon-Induced Hot-Electron TransferReviewers' Comments:

Reviewer #1:

Remarks to the Author:

The authors make very nice metal nanocluster-semiconductor epitaxial junctions. The physics of such junctions on a macroscale are well known for more than 100 years, and are an integral part of a multibillion dollar semiconductor device industry. The authors can consult the following website for the details: https://en.wikipedia.org/wiki/Metal-semiconductor_junction

That said, fortunately this industry does not yet depend on plasmonics. Nevertheless, The main point of this manuscript is that electron transfer from Ag to chalcogenide nanocrystals happens on 18 and 7.5 fs time scale as measured with 35 fs laser pulses. No data is shown that would confirm that this is actually measured, since all the time resolved data is shown on ps time scale. Moreover, the authors invoke the PICTT process, which I expect happens within the laser pulse, and thus does not have a time delay.

While I find the nanocrystals to be beautiful and their characterization to be rigorous, I find the electronic structure and time dependent measurements to be wanting. First of all, upon forming a metal semiconductor junction, one necessarily forms an interface state.¹ The way this manuscript is written, metals and semiconductors are described as independent entities, and their interface structures, which could define the putative ultrafast, high efficiency transfer, are completely ignored. It cannot be published this way. Moreover, the authors invoke that surface plasmon polaritons absorb the optical energy, and their decay generates hot electrons that are somehow injected into the semiconductors. These are certainly not surface plasmon polariton modes, because such excitations, to be defined, need to propagate on their wavelength scale on a metal surface. For 13 nm Ag particles, these must be the localized particle plasmon modes.

Finally, the authors seem to imply that the fast electron transfer to semiconductor domains is exceedingly fast. They are apparently not familiar that such fast time scales are predicted for Au nanoclusters on TiO₂,² and reported for Ag on TiO₂ and HOPG.⁴

Finally, finally, when the authors report a transient absorption spectrum they should assign it. If hot electron is transferred to the conduction band of a chalcogenide, and the hole remains in Ag, what is the absorption spectrum of this entity?

1 Wager, J. F. & Kuhn, K. Device Physics Modeling of Surfaces and Interfaces from an Induced Gap State Perspective. *Critical Reviews in Solid State and Materials Sciences* 42, 373-415, doi:10.1080/10408436.2016.1223013 (2017).

2 Long, R. & Prezhdo, O. V. Instantaneous Generation of Charge-Separated State on TiO₂ Surface Sensitized with Plasmonic Nanoparticles. *J. Am. Chem. Soc.* 136, 4343-4354, doi:10.1021/ja5001592 (2014).

3 Tan, S. et al. Plasmonic coupling at a metal/semiconductor interface. *Nature Photon* 11, 806-812, doi:10.1038/s41566-017-0049-4 (2017).

4 Tan, S. et al. Ultrafast Plasmon-Enhanced Hot Electron Generation at Ag Nanocluster/Graphite Heterojunctions. *J. Am. Chem. Soc.* 139, 6160-6168, doi:10.1021/jacs.7b01079 (2017).

Reviewer #2:

Remarks to the Author:

In the manuscript "Epitaxial Growth of Highly Symmetrical Branched Noble Metal-Semiconductor Heterostructures with Efficient Plasmon-Induced Hot-Electron Transfer", the authors describe the growth of CdS-Ag (and CdSe-Ag) nanostructures that have high hot-electron transfer quantum yields

of ~18%. The nanostructures consist of Ag icosahedral cores with twenty CdS nanorods epitaxially grown off the (111) facets of the Ag nanoparticles. Although similarly high plasmonic hot-electron transfer quantum yields have been previously reported in other kinds of samples, this work is noteworthy because of the well-controlled epitaxial growth of the complex nanostructure despite the large lattice mismatch. Extensive work went into the characterization of the sample chemical structure (TEM, HAADF-STEM, XRD, and XPS) and the data appears technically sound.

The group used mid-IR transient absorption spectroscopy to quantify the charge transfer efficiency. This technique directly probes for the presence of electrons in the conduction band of the CdS or CdSe and is the appropriate technique to quantify the charge transfer efficiency. Appropriate controls and calibration samples were used to determine the overall electron transfer efficiency.

Overall, the paper is broadly interesting to the plasmonics community and well written. I only have a couple of minor suggestions:

1. The authors provided a significant amount of characterization on the CdS-Ag icosapod structure, but the yield of the process was not discussed. For the growth of the CdS nanorods on the Ag interfaces, can the authors comment on the yield of the target Ag-CdS icosapod structures? Are other structures formed as well?

2. In previous work by Lian's group (Ref. 53), they measured a very high electron transfer QY of ~24% in CdSe/Au nanostructures where the charge transfer was also attributed to the PICTT mechanism. For that work, there was a single interface between a AuNP and a CdSe nanorod, in contrast to this work where the Ag is 3D surrounded by the CdS or CdSe. Yet, here a slightly lower QY was measured for the CdSe/Ag structure compared to the CdSe/Au structure. I find this surprising since one might expect the greater interfacial surface area would improve the QY. It would help if the authors comment in the paper about why this might be.

Reviewer #3:

Remarks to the Author:

The manuscript by Zhai et al. reports an epitaxial growth of semiconductors on noble metal templates to construct a complex noble metal-semiconductor heterostructure. The well-defined epitaxial interfaces between the noble metal and the semiconductor are revealed by Cs-corrected HAADF-STEM. An ultrafast plasmon-induced charge transfer process with remarkably high quantum yield is observed at the epitaxial Ag-CdS and Ag-CdSe interfaces. This highly interesting work could inspire further studies on materials synthesis and plasmon science. Therefore, I recommend this manuscript be published in Nature Communications after minor revisions.

1. The authors only used Ag nanocrystals with icosahedral structure as the templates. Could Ag nanomaterials with other morphologies be used to obtain noble metal-semiconductor heterostructures with different symmetries?

2. The models of the Ag-CdS/CdSe interface structures (Supplementary Figs. 13 and 14) are confusing. Why the structures of CdS and CdSe are emphasized here? Can the authors provide a detailed 3D model to illustrate the epitaxial interfaces between Ag and CdS/CdSe?

3. For the plasmon-induced charge transfer process in Ag-CdS and Ag-CdSe, the authors only used blank CdS and CdSe nanorods as the references (Figure 3e). What is the result if the mixture of pre-synthesized Ag nanocrystals and CdS nanorods is pumped by the 550 nm laser? Would the plasmon energy also be transferred from Ag nanocrystals to CdS nanorods? If so, is there any difference in the quantum yield of plasmon-induced hot electrons between epitaxial Ag-CdS heterostructures and the mixture of Ag nanocrystals and CdS nanorods?

4. The formats of some references are not corrected. For instance, "Science 353 (2016)" in Reference 39 should be "Science 353, aac5523 (2016)".

Our Response to Reviewers' Comments

Reviewer #1

Comment 1:

The authors make very nice metal nanocluster-semiconductor epitaxial junctions. The physics of such junctions on a macroscale are well known for more than 100 years, and are an integral part of a multibillion dollar semiconductor device industry. The authors can consult the following website for the details: https://en.wikipedia.org/wiki/Metal&semiconductor_junction.

That said, fortunately this industry does not yet depend on plasmonics. Nevertheless, the main point of this manuscript is that electron transfer from Ag to chalcogenide nanocrystals happens on 18 and 7.5 fs time scale as measured with 35 fs laser pulses. No data is shown that would confirm that this is actually measured, since all the time resolved data is shown on ps time scale. Moreover, the authors invoke the PICTT process, which I expect happens within the laser pulse, and thus does not have a time delay.

While I find the nanocrystals to be beautiful and their characterization to be rigorous, I find the electronic structure and time dependent measurements to be wanting.

Response 1: We thank the Reviewer very much for your invaluable comments. After carefully reading all the comments, we have carried out substantial and new experiments, which allows us to provide more comprehensive supportive evidence to address them. The revised parts are written in **red** in our revised manuscript. Our point-to-point responses are listed below.

First, we have to point out that the main point of our work is to develop a novel wet-chemical method for the preparation of hierarchical noble metal-semiconductor heterostructures with well-defined compositions, morphologies, and epitaxial interfaces. The developed synthetic strategy could provide significant guidance for the preparation of other novel heterostructures composed of different components with large lattice mismatches. Moreover, we found that the construction of an epitaxial noble metal-semiconductor interface would be a promising way to facilitate the plasmon-induced hot-electron transfer from noble metal to semiconductor.

Second, the time constants for the electron transfer from Ag to the CdS nanorods are on a time scale that occurs within the laser pulse, but the fit was convoluted with the instrument response function. Therefore, we can extract these time constants with some certainty. We have included zoomed-in insets of the kinetics for the Ag-CdS samples in

Fig. 3e. The error associated with the electron transfer rate is 18 ± 0.3 fs (inset in Fig. 3e). Similar to the previous works (*J. Am. Chem. Soc.* **136**, 4343-4354 (2014); *Science* **349**, 632-635 (2015); *Nat. Photon.* **11**, 806-812, (2017); *J. Am. Chem. Soc.* **139**, 6160-6168 (2017)), on such a short timescale, the plasmon decays directly into the electron in the conduction band (CB) of semiconductors before the rapid cooling of hot electrons induced by electron-electron scattering. We have added discussion about it in the **Ultrafast plasmon-induced charge transfer** section.

Fig. 3. (e) Time-resolved mid-IR absorption spectrum of Ag-CdS icosapods. Grey curves represent the multiexponential fit as described in Eq. S5. A negligible intraband absorption signal (black squares) is observed in a control experiment conducted on CdS nanorods. Inset in **e**: the zoomed-in kinetics spectrum for the Ag-CdS icosapods.

Revision:

The following discussion has been added in the **Ultrafast plasmon-induced charge transfer** section on Page 12 in the revised manuscript.

“The plasmon-induced hot-electron transfer processes in the epitaxial Ag-CdS and Ag-CdSe icosapods are much faster than the conventional indirect plasmon-induced hot-electron transfer process (commonly more than 100 fs⁵³⁻⁵⁵). Similar to previous reports, on such a short timescale (< 20 fs), the plasmon directly decays into the electrons in the CB of semiconductors *via* the interfacial charge-transfer transition before rapid cooling of hot electrons induced by electron-electron scattering^{54,55,58-60}. Unfortunately, because the transient IR selectively probes the electrons injected into the CB of CdS, similar to the previous reports⁵⁴, our study was not able to provide direct information about the effect of the metal-semiconductor interfacial states on the plasmon hot electron transfer and recombination processes.”

Comment 2:

First of all, upon forming a metal semiconductor junction, one necessarily forms an interface state.¹ The way this manuscript is written, metals and semiconductors are described as independent entities, and their interface structures, which could define the putative ultrafast, high efficiency transfer, are completely ignored. It cannot be published this way.

Response 2: We highly appreciate your invaluable comments. Based on your comments, we have used ultraviolet photoelectron spectroscopy (UPS) to study the interfacial electronic structure of the Ag-CdS icosapods.

As shown in **Supplementary Fig. 7a**, the cut-off energy of Ag nanocrystals and Ag-CdS icosapods is determined to be 16.95 eV and 17.20 eV, respectively. The corresponding work functions (Φ) can be estimated according to the following equation: $\Phi = h\nu - |E_{\text{cut-off}} - E_f|$, where $h\nu$ was the fixed incident photon energy of 21.22 eV (He I lamp) and E_f was calibrated to 0 eV using a standard Au sample. Φ can be estimated to be 4.27 eV for Ag nanocrystals and 4.02 eV for Ag-CdS icosapods. Here, the Fermi level of Ag nanocrystals and Ag-CdS icosapods should be -4.27 eV and -4.02 eV relative to the vacuum level, respectively. The cut-off energy of CdS nanorods is determined to be 17.30 eV. Therefore, the Fermi level of CdS should be -3.92 eV relative to the vacuum level (**Supplementary Fig 7b**). The distance between Fermi level and valence band (VB) of CdS is measured to be 2.30 eV by the linear fitting of the UPS spectrum in the long tail (**inset in Supplementary Fig. 7b**), demonstrating the valence band (VB) of CdS nanorods is -6.22 eV. Meanwhile, the bandgap of CdS nanorods can be obtained from bleach signal of CdS with center located at ~ 475 nm (2.61 eV) in the ultrafast pump-probe TA spectrum (**Supplementary Fig. 7c**). Therefore, the conduction band (CB) of CdS should be -3.61 eV. Based on the aforementioned results, the interfacial electronic structure could be schematically shown in **Supplementary Fig. 7d**. The shift (~ 0.25 eV) of the Fermi level in Ag-CdS icosapods compared with that of Ag nanocrystals demonstrates a Schottky barrier was formed at the Ag-CdS interface (*Crit. Rev. Solid State Mater. Sci.* **42**, 373-415 (2017)).

Supplementary Fig. 7. (a) UPS spectra of Ag nanocrystals and Ag-CdS icosapods. Inset: the zoom-in UPS spectra of Ag nanocrystals and Ag-CdS icosapods. (b) UPS spectra of CdS nanorods. Inset: the zoom-in UPS spectra of CdS nanorods. (c) Two-dimensional pseudo-color plots of pump-visible probe TA spectra of the colloid solution of CdS in toluene pumped by 400 nm laser. (d) The schematic illustration of the interfacial Schottky barrier in the Ag-CdS icosapods.

Revision:

In the **Heterostructure synthesis and structure characterizations** section on Page 8 in the revised manuscript, we have added the following discussion to illustrate the interfacial electronic structure in Ag-CdS icosapods.

“Ultraviolet photoelectron spectroscopy (UPS) was used to reveal the interfacial electronic structure of Ag-CdS icosapods. As shown in Supplementary Fig. 7a, the cut-off energies ($E_{\text{cut-off}}$) of Ag nanocrystals and Ag-CdS icosapods are determined to be 16.95 eV and 17.20 eV, respectively. The Fermi levels of Ag nanocrystals and Ag-CdS icosapods are estimated to be -4.27 eV and -4.02 eV relative to the vacuum level, respectively. Compared with the Ag nanocrystals, the shift ($\sim 0.25 \text{ eV}$) of the Fermi level of Ag-CdS icosapods demonstrates a Schottky barrier was formed between Ag

and n-type semiconductor CdS in the Ag-CdS icosapods⁵¹. Since the Fermi level, conduction band (CB), and valence band (VB) of CdS are measured to be -3.92 eV, -3.61 eV, and -6.22 eV, respectively (Supplementary Figs. 7b and c), the interfacial electronic structure of Ag-CdS icosapods can be schematically shown in Supplementary Fig. 7d.”

We have also added the following sentence on pg 12 of the **Ultrafast plasmon induced charge transfer** section:

“Unfortunately, because the transient IR selectively probes the electrons injected into the CB of CdS, similar to the previous reports⁵⁴, our study was not able to provide direct information about the effect of the metal-semiconductor interfacial states on the plasmon hot electron transfer and recombination processes.”

Comment 3:

Moreover, the authors invoke that surface plasmon polaritons absorb the optical energy, and their decay generates hot electrons that are somehow injected into the semiconductors. These are certainly not surface plasmon polariton modes, because such excitations, to be defined, need to propagate on their wavelength scale on a metal surface. For 13 nm Ag particles, these must be the localized particle plasmon modes.

Response 3: Thanks for this valuable suggestion. We have replaced SPR with “**localized surface plasmon resonance (LSPR)**”, and changed the schematic illustration of the plasmon-induced hot-electron transfer process in Ag-CdS icosapods in Fig. 3a in the revised manuscript.

Fig. 3. Plasmon-induced charge separation in Ag-CdS icosapods. (a) Schematic illustration of plasmon-induced hot-electron transfer in Ag-CdS icosapods after femtosecond laser pumping.

Comment 4: Finally, the authors seem to imply that the fast electron transfer to semiconductor domains is exceedingly fast. They are apparently not familiar that such

fast time scales are predicted for Au nanoclusters on TiO₂ and reported for Ag on TiO₂ and HOPG.

Response 4: Thanks for this valuable comment. Similar to these previous works (*J. Am. Chem. Soc.* **136**, 4343-4354 (2014); *Nat. Photon.* **11**, 806-812, (2017); *J. Am. Chem. Soc.* **139**, 6160-6168 (2017)), on such a short timescale, the plasmon decays directly into the electron in the CB of semiconductors before the rapid cooling of hot electrons induced by electron-electron scattering. Meanwhile, it should be noted that the size of plasmonic noble metal nanoparticles in our Ag-CdS icosapods (13.3 ± 0.6 nm of Ag) is quite different from that of Ag clusters/TiO₂, Au clusters/TiO₂, and Ag clusters/graphite systems (less than 2 nm of Ag or Au). Previous reports have demonstrated that the plasmon-induced hot-electron transfer efficiency in noble metal-semiconductor heterostructures decreases greatly at larger noble metal particle sizes (especially larger than 10 nm). With the increase in the size of noble metal nanoparticles, the noble metal nanoparticles could absorb and scatter more light and generate stronger near-field intensities. However, surface plasmons tend to decay by photon emission rather than by exciting electron-hole pairs. Therefore, the decay of surface plasmons in larger nanoparticles produces a much lower yield of highly energetic hot electrons, leading to less hot electrons with sufficient energy to cross the Schottky barrier into semiconductors (*ACS Nano* **10**, 957-966 (2016); *ACS Nano* **13**, 13610-13614 (2019)). For instance, the QY is much less than 1% in Au-tipped CdS nanorods when the Au nanoparticle diameter is larger than 6 nm (*Nano Lett.* **20**, 4322-4329 (2020)). Differently, high QY (~18.1%) has been achieved in our epitaxial Ag-CdS icosapods with large Ag nanoparticles (size of 13.3 ± 0.6 nm), indicating the uniqueness of our epitaxial noble metal-semiconductor interfaces. We have carefully compared them with our work and added some discussion in the revised manuscript (please see the following **Revision** part).

Revision:

The following discussion is added in the **Ultrafast plasmon-induced charge transfer** section on Page 12 in our revised manuscript.

“The plasmon-induced hot-electron transfer processes in the epitaxial Ag-CdS and Ag-CdSe icosapods are much faster than the conventional indirect plasmon-induced hot-electron transfer process (commonly more than 100 fs⁵³⁻⁵⁵). Similar to previous reports, on such a short timescale (< 20 fs), the plasmon directly decays into the electrons in the CB of semiconductors *via* the interfacial charge-transfer transition before rapid cooling of hot electrons induced by electron-electron scattering^{54,55,58-60}. Unfortunately, because the transient IR selectively probes the electrons injected into the CB of CdS, similar to

the previous reports⁵⁴, our study was not able to provide direct information about the effect of the metal-semiconductor interfacial states on the plasmon hot electron transfer and recombination processes.”

Comment 5: Finally, finally, when the authors report a transient absorption spectrum they should assign it. If hot electron is transferred to the conduction band of a chalcogenide, and the hole remains in Ag, what is the absorption spectrum of this entity?

Response 5: We thank the Reviewer for this comment. Unfortunately, we are unable to determine its spectrum due to the continuous energy band structure of the metal nanoparticle. In comparison to numbers of electrons mobile throughout the structure, the change in the spectrum would be indistinguishable due to such a small perturbation in the hole generation.

Reviewer #2

Comment 1:

In the manuscript “Epitaxial Growth of Highly Symmetrical Branched Noble Metal-Semiconductor Heterostructures with Efficient Plasmon-Induced Hot-Electron Transfer”, the authors describe the growth of CdS-Ag (and CdSe-Ag) nanostructures that have high hot-electron transfer quantum yields of ~18%. The nanostructures consist of Ag icosahedral cores with twenty CdS nanorods epitaxially grown off the (111) facets of the Ag nanoparticles. Although similarly high plasmonic hot-electron transfer quantum yields have been previously reported in other kinds of samples, this work is noteworthy because of the well-controlled epitaxial growth of the complex nanostructure despite the large lattice mismatch. Extensive work went into the characterization of the sample chemical structure (TEM, HAADF-STEM, XRD, and XPS) and the data appears technically sound.

The group used mid-IR transient absorption spectroscopy to quantify the charge transfer efficiency. This technique directly probes for the presence of electrons in the conduction band of the CdS or CdSe and is the appropriate technique to quantify the charge transfer efficiency. Appropriate controls and calibration samples were used to determine the overall electron transfer efficiency.

Overall, the paper is broadly interesting to the plasmonics community and well written. I only have a couple of minor suggestions.

Response 1: We thank the Reviewer very much for the positive comments on our work. Based on the reviewers’ invaluable comments, we have carried out substantial and new experiments, which allows us to provide more comprehensive supportive evidence to address them. The revised parts are written in red in our revised manuscript. Our point-to-point responses are listed below. The point-by-point responses are given below.

Comment 2:

The authors provided a significant amount of characterization on the CdS-Ag icosapod structure, but the yield of the process was not discussed. For the growth of the CdS nanorods on the Ag interfaces, can the authors comment on the yield of the target Ag-CdS icosapod structures? Are other structures formed as well?

Response 2: We thank the reviewer for raising the questions. The synthetic method we developed is robust. The yield of the target Ag-CdS icosapods structures is ~100% using Ag icosahedral nanocrystals with narrow size distribution. As shown in the TEM image of hundreds of Ag-CdS icosapods (**Fig. R1**), no other structures can be observed.

Fig. R1. TEM image of hundreds of Ag-CdS icosapods. (Scale bar: 100 nm).

Comment 3:

In previous work by Lian's group (Ref. 53), they measured a very high electron transfer QY of ~24% in CdSe/Au nanostructures where the charge transfer was also attributed to the PICTT mechanism. For that work, there was a single interface between a AuNP and a CdSe nanorod, in contrast to this work where the Ag is 3D surrounded by the CdS or CdSe. Yet, here a slightly lower QY was measured for the CdSe/Ag structure compared to the CdSe/Au structure. I find this surprising since one might expect the greater interfacial surface area would improve the QY. It would help if the authors comment in the paper about why this might be.

Response 3: We thank the reviewer for this valuable comment. Notably, the sizes of plasmonic noble metal nanoparticles in our Ag-CdS icosapods and the Au-tipped CdSe nanorods in Ref. 53 are quite different (4 nm of Au in Au-tipped CdSe nanorods and 13.3 ± 0.6 nm of Ag in Ag-CdS icosapods). Previous reports have demonstrated that plasmon-induced hot-electron transfer efficiency in noble metal-semiconductor heterostructures would decrease greatly when particle size becomes larger. With the increase in the size of noble metal nanoparticles, surface plasmons tend to decay by photon emission rather than by exciting electron-hole pairs, leading to fewer hot electrons having sufficient energy to cross the Schottky barrier into semiconductors

(*ACS Nano* **10**, 957-966 (2016); *ACS Nano* **13**, 13610-13614 (2019)). For instance, the QY of the plasmon-induced hot-electron transferred from noble metal to semiconductor would be less than 1% in Au-tipped CdS nanorods when the diameter of Au nanoparticle is larger than 6 nm (*ACS Nano* **10**, 957-966 (2016); *Nano Lett.* **20**, 4322-4329 (2020)). Till now, high QY of the plasmon-induced hot-electron transferred from noble metal to semiconductor could only achieve in heterostructures composed of tiny noble metal clusters (less than 4 nm) and semiconductors (*Science* **349**, 632-635 (2015); *Nat. Photon.* **11**, 806-812, (2017); *J. Am. Chem. Soc.* **139**, 6160-6168 (2017); *Nano Lett.* **20**, 4322-4329 (2020)). However, high QYs (~18.1%) have been achieved in our Ag-CdS icosapods with large Ag nanoparticles (size of 13.3 ± 0.6 nm), indicating the superiority and significance of our epitaxial heterostructures.

Reviewer #3

Comment 1:

The manuscript by Zhai *et al.* reports an epitaxial growth of semiconductors on noble metal templates to construct a complex noble metal-semiconductor heterostructure. The well-defined epitaxial interfaces between the noble metal and the semiconductor are revealed by Cs-corrected HAADF-STEM. An ultrafast plasmon-induced charge transfer process with remarkably high quantum yield is observed at the epitaxial Ag-CdS and Ag-CdSe interfaces. This highly interesting work could inspire further studies on materials synthesis and plasmon science. Therefore, I recommend this manuscript be published in Nature Communications after minor revisions.

Response 1: We thank the Reviewer very much for the positive comments on our work. Based on the comments, we have carried out substantial and new experiments, which allows us to provide more comprehensive supportive evidence to address them. The revised parts are written in red in our revised manuscript. The point-to-point replies are given below.

Comment 2:

The authors only used Ag nanocrystals with icosahedral structure as the templates. Could Ag nanomaterials with other morphologies be used to obtain noble metal-semiconductor heterostructures with different symmetries?

Response 2: We thank the reviewer for this invaluable comment. Based on the comment, we have used octahedral single-crystalline Ag nanocrystals with eight exposed (111) facets as seeds (**Fig. R2**), and Ag-CdS heterostructures with eight CdS nanorods (Ag-CdS octapods) were obtained (**Fig. R3**). The representative HRTEM images of two Ag-CdS octapods viewed with C_2 (**Fig. R3b**) and C_3 (**Fig. R3c**) symmetries show four and six radial nanorods, respectively. Dark-field STEM image and the corresponding EDS elemental mapping of a single Ag-CdS octapod (**Fig. R3d**) viewed along its C_3 rotational axis show that the Cd and S distribute homogeneously on six nanorods and Ag element locates in the core area. Therefore, Ag nanomaterials with other morphologies can be used to obtain noble metal-semiconductor heterostructures with different symmetries.

Fig. R2. (a) TEM, (b) HRTEM images of synthesized single crystalline Ag nanocrystals. (Scale bars: 50 nm in a; 2 nm in b).

Fig. R3. (a) TEM, (b,c) HRTEM images of synthesized Ag-CdS octapods with (b) C_2 and (c) C_3 symmetry. (d) HAADF-STEM and the corresponding EDS elemental mapping of a single Ag-CdS octapod. (Scale bars: 50 nm in a; 20 nm in b to d).

Comment 3:

The models of the Ag-CdS/CdSe interface structures (Supplementary Figs. 13 and 14) are confusing. Why the structures of CdS and CdSe are emphasized here? Can the authors provide a detailed 3D model to illustrate the epitaxial interfaces between Ag and CdS/CdSe?

Response 3: We thank the reviewer for raising these questions. We have built a detailed 3D model of the Ag-CdS interfacial structures for better understanding. Please see the “**Interface Structure**” in the Supplementary Material. The following is the 2D projection of the 3D model of the Ag-CdS interface (**Fig. R4**).

Fig. R4. 2D projection of the 3D model of the Ag-CdS interface.

Comment 4:

For the plasmon-induced charge transfer process in Ag-CdS and Ag-CdSe, the authors only used blank CdS and CdSe nanorods as the references (Figure 3e). What is the result if the mixture of pre-synthesized Ag nanocrystals and CdS nanorods is pumped by the 550 nm laser? Would the plasmon energy also be transferred from Ag nanocrystals to CdS nanorods? If so, is there any difference in the quantum yield of plasmon-induced hot electrons between epitaxial Ag-CdS heterostructures and the mixture of Ag nanocrystals and CdS nanorods?

Response 4: We thank the reviewer for raising the questions. We have performed the experiments. Our result shows that after mixing Ag and CdS particles in a 1:1 ratio in solution and pumping at 550 nm, a negligible signal (less than 0.5 mOD) was observed (**Fig. R5c**). A similar magnitude of the signal can be seen in the Ag nanocrystals (**Fig. R5a**). Therefore, in the mixture of Ag and CdS, the signal observed could arise from the Ag itself. Here, we could conclude that no plasmon energy would be transferred from Ag nanocrystals to CdS nanorods in the physical mixture of Ag nanocrystals and CdS nanorods.

Fig. R5. Averaged mid-IR TA spectra of (a) Ag nanocrystals, (b) CdS nanorods, (c) the physical mixture of Ag nanocrystals and CdS nanorods after 550 nm laser excitation.

Comment 5:

The formats of some references are not corrected. For instance, “Science 353 (2016)” in Reference 39 should be “Science 353, aac5523 (2016)”.

Response 5: We have checked and corrected the formats of some references. We sincerely thank the reviewer for the careful reading of our manuscript.

Reviewers' Comments:

Reviewer #1:

Remarks to the Author:

The authors have revised their manuscript substantially to respond to my comments. I do not for a moment believe the rise times they report, but that is not the main point. I recommend publication.

Reviewer #2:

Remarks to the Author:

The authors have addressed the concerns of the reviewers, and the manuscript is sound.

Reviewer #3:

Remarks to the Author:

The authors have fully addressed my questions in the revised manuscript. I would recommend publication of the manuscript in its current form.

Our Response to Reviewers' Comments

Reviewer #1

Comment 1: The authors have revised their manuscript substantially to respond to my comments. I do not for a moment believe the rise times they report, but that is not the main point. I recommend publication.

Response 1: We thank the Reviewer very much for the positive comment.

Reviewer #2

Comment 1: The authors have addressed the concerns of the reviewers, and the manuscript is sound.

Response 1: We thank the Reviewer very much for the positive comment.

Reviewer #3

Comment 1: The authors have fully addressed my questions in the revised manuscript. I would recommend publication of the manuscript in its current form.

Response 1: We thank the Reviewer very much for the positive comment.